# Sleep and Safety among Healthcare Workers: The Effect of Obstructive Sleep Apnea and Sleep Deprivation on Safety

**DOI:** 10.3390/medicina58121723

**Published:** 2022-11-24

**Authors:** Likhita Shaik, Mustafa S. Cheema, Shyam Subramanian, Rahul Kashyap, Salim R. Surani

**Affiliations:** 1Hennepin Healthcare, Minneapolis, MN 55415, USA; 2G9QC+GGM CMH Lahore Medical College, Abdul Rehman Rd, Sarwar Colony, Lahore 54000, Pakistan; 3Sutter Gould Medical Foundation, Tracy, CA 95376, USA; 4Well Span Health, York, PA 17403, USA; 5Department of Pulmonary, Critical Care & Sleep Medicine, Texas A&M University, College Station, TX 77843, USA

**Keywords:** OSA, sleep disorders, SD, healthcare professionals, patient safety, medical errors

## Abstract

Almost one billion people worldwide are affected by Obstructive Sleep Apnea (OSA). Affected individuals experience disordered breathing patterns during sleep, which results in fatigue, daytime drowsiness, and/or sleep deprivation. Working under the influence of these symptoms significantly impairs work productivity and leads to occupational accidents and errors. This impact is seen in healthcare workers (HCWs) who are not immune to these conditions. However, poorly controlled OSA in this subset of individuals takes a heavy toll on patient care due to the increased risk of medical errors and can also alter the mental and physical well-being of the affected HCW in various ways. OSA and safety issues have been recognized and mitigated among the airline and transport industries; however, the healthcare industry lags in addressing these concerns. This article reviews hypersomnolence and sleep disorder as key clinical features of OSA and their effect on HCW safety.

## 1. Introduction

Obstructive sleep apnea (OSA) is a sleep-related breathing disorder that causes repetitive partial or complete obstruction of the upper airways during sleep [1]. It affects approximately 25% of the global population, and the United States ranks among the countries with the highest prevalence rates of OSA (21.6%). The prevalence of OSA is higher among males than females (25% vs. 13%) [1,2] and many OSA cases remain underdiagnosed or undiagnosed [3]. Any patient affected by OSA experiences the negative effects of apnea–hypopnea episodes through poor quality of overnight sleep, excessive daytime sleepiness (EDS), and associated symptoms such as fatigue, lack of concentration, and poor work efficiency. Though OSA affects individuals across occupations, when poorly controlled OSA is observed in healthcare workers (HCWs) it can adversely affect the safety of the HCWs and patients because of sleep deprivation (SD) and EDS symptoms. However, there is an absence of studies that address these serious concerns with potential solutions [4,5,6,7]. In other industries—for example, the airline and transportation industry—there are strict requirements and screening methods in place. In order for an employee with OSA to be able to work, they must show efficacy and compliance with therapy. The healthcare industry has not adopted screening measures and compliance with therapy policies to date. Our article provides an overview of sleep problems among healthcare providers, the impact of these problems on safety in the healthcare environment, and important clinical implications for improving outcomes. It is important to note that data pertaining to the prevalence of OSA among HCWs are sparse, and more work needs to be conducted in this area. With long shifts being common in HCW environments, the lack of these measures and policies leads to safety concerns.

## 2. Review of the Literature

### 2.1. Risk Factors for Sleep Morbidity

Obesity, advanced age, male sex, neck circumference (17 inches and 16 inches for men and women, respectively), smoking, craniofacial abnormalities, and a family history of OSA are the most common risk factors for developing OSA. Obesity remains a major modifiable risk factor for OSA [1]. Lack of physical activity is another emerging risk factor for OSA [8]. Moreover, with advances in technology and poor work–life balance, a large study by Mayne et al. emphasizes the growing concern regarding a sedentary lifestyle in HCWs [9]. Stress is another major component that has co-existing comorbidity in OSA, wherein the severity of stress correlates with poor health outcomes in OSA [10]. The healthcare industry is a high-stress environment which may lead to an inclination to develop habits such as smoking, alcohol usage, and physical inactivity, all of which, as mentioned earlier, can predispose an individual to OSA [9,10,11]. Furthermore, increased stress from pandemics has been associated with higher incidences of sleep-related disorders among physicians and nurses [12,13,14].

### 2.2. Impact on Sleep Duration and Quality 

Sleep is essential for the optimal functioning of the body and mind. The National Sleep Foundation in the United States recommends from 7 to 9 h and 7 to 8 h of sleep in young and older adults, respectively [15]. SD continues to be a rising concern for people all over the world. According to the Centre for Disease Control and Prevention’s (CDC) reports in 2016, 35.2% of all adults in the United States sleep <7 h on average per night [16]. Various mechanisms have been described that correlate poor sleep and OSA. Although the pathology of OSA frequently describes OSA as causing poor sleep quality, a new element in the pathology also correlates with the reverse. The frequent arousals from sleep at night due to obstructions in breathing can lead to inadequate sleep, and reduced sleep also decreases the activity of the muscle (genioglossus) that maintains the patency of the upper airways during sleep [17]. The OSA can obstruct the already narrow airway in OSA due to genioglossus hypoactivity, hence further worsening OSA. In another study, a lower minimum oxygen saturation after SD was seen in subjects with OSA compared to those without OSA [18]. All of these factors result in insufficient and poor sleep quality and can further exacerbate OSA [19,20]. The adversities of SD and other disturbances are well-reported among HCWs (Table 1) [21,22,23,24,25,26,27,28,29,30,31,32,33]. Several studies report the high burden of SD among HCWs, especially when combined with advanced age (>65 years), OSA, alcohol abuse, and clinical stressors such as shift work, research work, medical education, and family responsibilities [34,35]. A study in 2016 reported SD in 44.2% of young physicians, of which 40.4% were diagnosed with OSA [4]. Recent studies during the COVID-19 pandemic report a dramatic decline in sleep duration, with a mean duration of only 6.1 h per night and other sleep-related disturbances, leading to an increased usage of sedative drugs by HCWs [13,14]. In summary, in addition to the sleep-depriving effects of OSA, it can cause a vicious cycle of events that worsen sleep and are also related to OSA outcomes.

### 2.3. Impact of Sleep Morbidity in HCWs

Poorly controlled OSA can manifest in excess sleepiness, daytime fatigue, and impaired concentration, all of which can have significant safety implications, particularly in the healthcare setting [36].

In a study by Weaver et al., researchers noted that a positive screening result for sleep disorders such as OSA and insomnia was associated with a nearly fourfold increase in odds of physician burnout [37]. The odds of reporting a conflict with other professionals, alcohol use, sleep-altering medications use, and significant weight changes increased with sleep loss [13,14,38,39]. Papp et al. reported that resident physicians felt their interactions with other healthcare staff were negatively affected by sleep disturbance. Interest in teaching other residents was also reduced [25]. Likewise, physicians who slept five or fewer hours per night were 2.02 times more likely to have been named in a malpractice suit [40]. Guglielmi et al. conducted a survey to assess job stress, burnout, and job dissatisfaction in patients with OSA [41]. Lockley et al. reported that physicians-in-training who worked more than 24-h on-call shifts were at increased risk of sharp occupational injuries [4]. The reasons for this could be fatigue, increased workload, and lapses in concentration [42,43,44]. Patients with OSA reported high levels of emotional exhaustion [41,45]. Studies indicate that nearly one in two trainee physicians and practicing physicians have reported symptoms of burnout [46]. It is well known that burnout is linked to a variety of health issues, including depression, drug addiction, suicidal thoughts, and sleeplessness [46,47]. It is clear from these data that sleep morbidity in HCWs, including sleep deprivation, poor quality sleep, insomnia, and OSA, directly affects both their physical and mental health, and indirectly can have a significant impact on the patient’s safety.

### 2.4. Impact on Patient Care

Physicians operate in an environment requiring intense concentration as well as mental and physical endurance. Dealing with issues that have a direct influence on the lives of patients requires alertness and smart decision-making. At times, unexpected challenges with uncertain outcomes may be encountered. Effective communication among healthcare professionals is essential in such situations. In these circumstances, when a HCW with poorly controlled OSA works during a healthcare shift, their work productivity is impaired, which impacts their ability to provide patient care. Studies indicate a strong relationship between OSA and workplace accidents when compared to all other sleep disorders, with a relative odds ratio of 2.88 [48]. A systematic review and meta-analysis of occupational accidents in workers with OSA revealed that the odds of work accidents were nearly doubled in workers with OSA compared to those without OSA [49]. A cross-sectional study in Thailand and Malaysia showed that 40.4% of young physicians have OSA. The incidence of fatigue, SD, and perception of inadequate sleep amongst these individuals was 65.4%, 44.2%, and 61.5%, respectively [50]. Additionally, a cross-sectional study involving the screening of sleep disorders revealed that 40.9% of healthcare workers reported at least one type of sleep disorder [37]. SD is a prominent symptom of OSA, and the effects of SD are well known (Table 1). A survey revealed that over one-third of physicians felt sleep-deprived every week, impacting patient care and safety. This survey also showed that physicians suffered from several side effects of SD, such as poor concentration, impaired decision-making, and mental health issues. Over 25% of these physicians felt that fatigue impacted their ability to adequately treat patients [47]. In a study by Papp et al., many residents had concerns about sleep disturbances resulting in a greater number of medical errors, specifically errors in entering information in patient records and prescribing medications. At times, incorrect dosages were prescribed, while, in other situations, medications were prescribed to the wrong patients [25].

Emergencies may arise in hospitals and require a rapid response from HCWs. Patients may need to be rushed to the operating room quickly. Patients may develop unexpected complications during surgery and require rapid intervention. In such circumstances, HCWs such as paramedics, surgeons, and nurses with OSA symptoms such as SD, EDS fatigue, stress, and burnout may pose a challenge in such circumstances. A study on surgeons after a night-on call showed impaired speed and accuracy in simulated laparoscopic performance. Several factors, such as stress and emotional demand, may lead to these effects [51]. Another study revealed that after a group of surgeons was awake the whole night on call, they made 20% more errors and took 14% more time to complete procedures compared to another group with good sleep [31]. A study by Carvalho et al. reported a high incidence of EDS symptoms among HCWs. These symptoms were also associated with increased alcohol consumption in patients between 30 and 49 years of age [52]. Being awake for 24 h is similar to having a blood alcohol content (BAC) of 0.10%. This is higher than the legal limit of 0.08% BAC in the United States [53]. Physicians who worked more than 24 h on-call shifts had double the number of attentional failure events when working overnight. They also committed 36% more serious medical errors in comparison to those working fewer-hour shifts. This study also noted 300% more sleep- and fatigue-related medical errors resulting in patient deaths [4]. Additionally, reaction time slows down while risk-taking behavior increases [54,55,56,57]. Moreover, workers with OSA are about ten times more likely to have a workplace disability. This can substantially increase costs for employers and hiring organizations due to the reduction in work productivity and increase in healthcare costs and liability in the event of mishaps [58]. All these effects of OSA-related symptoms in HCW indicate poor patient care, a higher risk of medical errors, and the possibility of mortality events. However, minimal literature is available that directly correlates OSA symptoms to effects on patient care, which warrants further research.

### 2.5. Impact on the General Population

The safety risks related to OSA, specifically road safety and workplace safety, have been extensively described in the literature. The HCW is not immune to these safety hazards even if they have OSA or SD resulting from long duty hours. OSA has been the probable cause of 10 highway and rail accidents investigated by the National Transport & Safety Board (NTSB) in the past 17 years, and OSA is an issue being examined in several ongoing NTSB rail and highway investigations [59]. The literature revealed that drivers with <6 h of sleep are 33% more likely to have a road accident than those with a higher number of hours of sleep [4,5,6,7]. Physicians who are sleep-deprived are more likely to be involved in road traffic accidents due to drowsiness [60]. Likewise, physician residents are at increased risk of dangerous driving after working extended shifts of more than 16 h [7]. A study has reported that patients with OSA had a sevenfold greater rate of automobile accidents than subjects without sleep apnea [61]. Vakulin et al. compared the effects of sleep restriction and alcohol on driving simulator performance in patients with OSA [62]. Patients with OSA were more likely to have had at least one accident from falling asleep while driving [63]. Furthermore, another study reported there could be a significant improvement in the driving performance of patients with OSA after successful treatment of their apnea with nasal Continuous Positive Airway Pressure (CPAP) [64]. Moreover, compliance with the CPAP is good for patients with moderate and severe OSA [65]. Another study by Karimi et al. reported that patients with sleep apnea were from 2.3 to 2.6 times more likely to be involved in a motor vehicle accident while driving compared with a control group of drivers without OSA [63]. Moreover, predictors of increased risk of a motor vehicle accident in OSA patients included a short sleep duration of 5 h or less, excessive sleepiness during the daytime, and using sleeping pills [63]. Although OSA puts the public at risk of such dangerous road incidents, no direct research is available to examine the accident rate caused by HCWs in OSA patients. However, with the available literature, increased accident rates due to OSA symptoms in HCWs would not be surprising.

### 2.6. Occupational Safety

In the aviation industry, OSA is considered a safety risk and a source of accidents [65]. Pilots who screen positively for OSA are initially disqualified and must seek treatment for OSA, along with the issuance of a special medical certificate, before flying [66,67]. In order for pilots to continue flying, physicians treating pilots with OSA are required to report whether the ongoing treatment is effective [68]. Federal motor carrier safety administration disqualifies a Commercial Motor Vehicle (CMV) driver with moderate to severe sleep apnea. They should not drive if they are not being treated. Complying with the treatment norms offers the most reliable way to secure an existing job for a commercial driver with sleep apnea. This increases the an individual’s ability to do his or her job safely at higher levels of alertness without compromising public safety.

Other occupations such as law enforcement establish medical standards to identify evidence of hypersomnolence or drowsiness during waking hours. The Association of American Railroads, an industry group, said railroads are continuing to take steps to combat worker fatigue, including confidential sleep disorder screening and treatment [69,70]. Late last year, the Federal Railroad Administration (FRA) issued a safety recommendation for railroads to screen and test for sleep apnea given the dangers of OSA in everyday life [71,72]. However, no rules or regulations exist to protect patient safety due to hidden but adverse events of OSA in HCWs. As discussed, HCWs are predisposed to an increased risk of developing OSA and facing the consequences of its worsening. It is unfair to expect a HCW with OSA and SD to perform similarly compared to any other HCW without OSA and SD. Quantifying this risk may be essential to realize the burden of the situation through further research. Moreover, there are no clear-cut-offs to define OSA diagnostic and treatment standards for HCWs in order to make them eligible to work and provide a safe patient care environment.

## 3. Clinical Implications


Healthcare professionals are at increased risk of developing OSA and its worsening effects. However, further research needs to be conducted to validate these concerns.OSA and its effects on patient care is an avenue for further research.As part of the medical assessment in the workplace, there is an immediate requirement to develop and implement improved strategies to screen and manage OSA in workers.Symptoms of OSA can be debilitating to HCWs and their families. Relaxation of existing stringent work hours and shift schedules and offering an allowance for the treatment of OSA can be rewarding and decrease the burden of psychological symptoms in these HCWs.HCWs with OSA should not be allowed to drive after a night sleep-deprived equivalent shift. Incentives should be offered to encourage adequate treatment of OSA.Healthcare providers should be made aware of the hidden effects of this disease in HCW to be able to address all concerns revolving around OSA in these patients that could directly or indirectly impact the disease progression.


## 4. Conclusions

The widespread prevalence of OSA and sleep deprivation in HCWs needs to be taken seriously, as these issues can significantly impact the work of the healthcare professionals who suffer from it. Research and data in this area are lacking among HCWs. SD, EDS, fatigue, and burnout can be the consequences of OSA, which, in HCWs, can be detrimental to patient care if not well controlled. These effects compound to reduce the overall efficiency and productivity of healthcare professionals. As a result, medical errors occur, leading to poor patient outcomes. A formal assessment of the effects of OSA on HCW health and patient care through research is necessary to substantiate its real burden. Screening HCWs for OSA and establishing work guidelines for OSA-affected HCWs and the necessary treatment that must be followed should be a standard protocol in all healthcare settings.

## Figures and Tables

**Table 1 medicina-58-01723-t001:** Adversities of SD in healthcare settings.

Study	Population (*n* = Sample Size)	Findings
Effect of on-call-related sleep deprivation on physicians’ mood and alertness [21]	Physicians (100)	10% of male participants reported experiencing a car accident while driving home after working an on-call shift. Of all participants, the percentage of alert physicians post-on-call was significantly reduced compared to the percentage pre-on-call.
Assessment of Physician Sleep and Wellness, Burnout, and Clinically Significant Medical Errors [22]	Attending physicians and house staff physicians (11,395)	Sleep loss and burnout can lead to medical errors and patient harm by 37.7% of attending physicians and 39.9% of trainee physicians.
The Association of Sleep Deprivation on the Occurrence of Errors by Nurses Who Work the Night Shift [23]	Nurses (138)	Medical errors committed by sleep-deprived nurses were significantly increased when compared to the means of medical errors by non-sleep-deprived nurses, *p* < 0.003.
Sleep loss and performance in residents and nonphysicians: a meta-analytic examination [24]	Physicians (959), non-physicians (1028), individual effect indexes (5295)	60 studies on SD showed that even <30 h of sleep loss decreased physicians’ overall performance by approximately 1 standard deviation and clinical performance by >1.5 standard deviations, resulting in increased medical errors.
The effects of sleep loss and fatigue on resident physicians: a multi-institutional, mixed-method study [25]	Residents (149)	Due to lack of sleep, residents described instances of falling asleep while talking to patients, writing patient notes, dictating, and reviewing the investigation.
The Intern and Sleep Loss [26]	Residents (not available)	The mood was significantly affected by sleep loss, and the affected individuals felt increased sadness, and decreased vigor, egotism, and social affection.
Stresses affecting surgical performance and learning [27]	Surgeons, senior residents, junior residents, interns, and medical students (33 operative procedures)	The performance of the individuals with 2 h of sleep was inferior when compared to those residents with normal sleep. They termed this ‘operative inefficiency,’ marked by indecision and poorly planned maneuvers exceeding 30% of the operating time.
The psychological impact of the COVID-19 pandemic and burnout severity in French residents: A national study [28]	Residents (1050)	This prospective study showed that interns committed more serious medical errors in Intensive Care Units (ICU) when working frequent shifts of 24 h or longer compared to those working shorter shifts.
Effect of reducing interns’ work hours on serious medical errors in intensive care units [29]	Residents (20)	These physicians who worked more than 24-h on-call shifts had twice as many attentional failures when working overnight and committed 36% more serious medical errors compared to those working 16-h shifts. This study also noted 300% more sleep- and fatigue-related medical errors resulting in patient deaths.
The Effects of Sleep Loss on Medical Residents’ Emotional Reactions to Work Events: a Cognitive-Energy Model [30]	Residents (78)	This study focused on the negative emotive effects of disruptive events while reducing the positive effect of goal-enhancing events related to sleep loss.
Effect of sleep deprivation on surgeons’ dexterity on laparoscopy simulator [31]	Surgeons in training (6)	Sleep-deprived surgeons made 20% more clinical errors and took 14% more time to complete tasks when compared to those without sleep deprivation.
Sleep Quality and Fatigue Among Prehospital Providers [32]	EMS professionals (119)	The mean sleep quality score of subjects experiencing severe fatigue at work was significantly lower than the mean sleep quality score among the non-fatigued subjects.
Lack of Sleep Symptoms in Healthcare Workers Leads to Safety Dangers [33]	CDC Surveillance data from 1995–1999	Sleep disturbances in HCWs cause them to: Commit 36% more serious medical errors than those whose scheduled work is limited to hours. Commit 5 times as many serious diagnostic errors. Have twice as many on-the-job attentional failures at night. Experience 61% more needlesticks and other sharp injuries after their 20th consecutive hour of work. Double their risk of a driving accident on their way home after long hours of work.

Adversity of SD in a healthcare setting.

## Data Availability

Not applicable.

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
