# Peer review of "Sleep and Safety among Healthcare Workers: The Effect of Obstructive Sleep Apnea and Sleep Deprivation on Safety"

_medicina, 2022, doi:10.3390/medicina58121723_

Round 1
Reviewer 1 Report (Previous Reviewer 1)
CPAP abbreviation is not expanded the first time it appears – it is expanded the second time it appears. Please discuss the factor affecting CPAP adherence as the topic is mentioned, see doi: 10.3390/jcm11010139.
Please add in the title that this is a narrative review.
Otherwise, the quality of the manuscript has been sufficiently improved.
Author Response
Reviewer 1
Comments and Suggestions for Authors
CPAP abbreviation is not expanded the first time it appears – it is expanded the second time it appears. Please discuss the factor affecting CPAP adherence as the topic is mentioned, see doi: 10.3390/jcm11010139.
Please add in the title that this is a narrative review.
It has been added the CPAP write out in the manuscript and reference has been added
Otherwise, the quality of the manuscript has been sufficiently improved.
We thank the reviewers for their diligence and their careful review of our manuscript – their insightful comments and feedback have enhanced the quality of our review, and we thank them for their time.

Reviewer 2 Report (Previous Reviewer 3)
Thank you for editing the manuscript as for the comment previously suggested. Overall, the soundness and the novelty of the article quite improved. However, a major shortcoming of the paper is that the third section is supposed to have only literature on OSA (as the title is "OSA in HCE s a safety concern for all"), whereas instead it mixes literature on sleep deprivation and OSA. See below my extensive comments and my suggestion on how to solve this inconsistency.
Major correction:
- as previously pointed out, in the section of "OSA in HCW is a safety concern for all", the studies reported should only address sleep loss and sleep deprivation connected to OSA, because the title of the section is OSA in HCW. If sleep deprivation, fatigue, sleep loss, or burnout are due to the loading on healthcare physicians and residents, this cannot be an evidence supporting the importance of screening for OSA. I see two alternatives to solve this issue: either eliminating all the references that do no investigate OSA in HCW, or creating a new section where the importance of sleep has on HCW, regardless of the medical condition of OSA. The same issue occurs in the section of "Impact of patient care". This should only indicate references where the sleep deprivation is due to OSA, otherwise we are assessing the sleep opportunity, and not poor sleep due to OSA. Either removing the references, or creating a subsection where you explain harmful consequences of sleep deprivation, and then you can state that sleep deprivation is peculiar of OSA, so all the above mentioned effects are enhanced with OSA condition.
- why "HCWs are predisposed to an increased risk of developing OSA?" I agree that they are predisposed to face consequences of its worsening, but are they really at high risk of OSA? If they are sleep deprived, this does not mean that they will develop OSA. I would like to hear the authors' thoughts and to see a reference that can adequately support this statement.
Minor correction:
- please provide a reference to the definition of OSA (line 33-35) and to the prevalence rate of OSA in the US (lines 36-37).
- provide reference to lines 66-67 (risk factors of OSA)
- avoid the & in the title section of "Impact on sleep duration and quality"(line 98)
- I would be careful in lines 144-146 stating that a poorly controlled OSA can result in cancer. This has been only recently hypothesized. Moreover, the 3 references quoted do not refer to cancer. I suggest including this reference, (doi: 10.1097/MD.0000000000028930) but I also encourage smoothening the strong statement.
Author Response
Thank you for editing the manuscript and for the comment previously suggested. Overall, the soundness and the novelty of the article quite improved. However, a major shortcoming of the paper is that the third section is supposed to have only literature on OSA (as the title is "OSA in HCE s a safety concern for all"), whereas instead, it mixes literature on sleep deprivation and OSA. See below my extensive comments and my suggestion on how to solve this inconsistency.
Major correction:
- as previously pointed out, in the section "OSA in HCW is a safety concern for all," the studies reported should only address sleep loss and sleep deprivation connected to OSA because the title of the section is OSA in HCW. If sleep deprivation, fatigue, sleep loss, or burnout are due to the loading on healthcare physicians and residents, this cannot be evidence supporting the importance of screening for OSA. I see two alternatives to solve this issue: either eliminating all the references that do no investigate OSA in HCW or creating a new section where the importance of sleep has on HCW, regardless of the medical condition of OSA. The same issue occurs in the section "Impact of patient care." This should only indicate references where sleep deprivation is due to OSA. Otherwise, we are assessing the sleep opportunity and not poor sleep due to OSA. Either remove the references or create a subsection where you explain the harmful consequences of sleep deprivation and then you can state that sleep deprivation is peculiar to OSA, so all the above-mentioned effects are enhanced with the OSA condition.
This section has been rewritten.
- why are "HCWs predisposed to an increased risk of developing OSA?" I agree that they are predisposed to face the consequences of its worsening, but are they really at high risk of OSA? If they are sleep deprived, this does not mean that they will develop OSA. I would like to hear the authors' thoughts and to see a reference that can adequately support this statement.
We agree that this is speculative, and hence we have removed this section.
Minor correction:
- please provide a reference to the definition of OSA (lines 33-35) and to the prevalence rate of OSA in the US (lines 36-37).
We have provided this.
- provide reference to lines 66-67 (risk factors of OSA)
This section has been re-written/edited
- avoid the & in the title section of "Impact on sleep duration and quality"(line 98)
This has been corrected
- I would be careful in lines 144-146, stating that a poorly controlled OSA can result in cancer. This has been only recently hypothesized. Moreover, the 3 references quoted do not refer to cancer. I suggest including this reference (doi: 10.1097/MD.0000000000028930), but I also encourage smoothening the strong statement.
This section has been revised, and this particular section has been removed.
We thank all the reviewers for their diligence, and careful review of our manuscript – their insightful comments and feedback have enhanced the quality of our review, and we thank them for their time.

Round 2
Reviewer 2 Report (Previous Reviewer 3)
Thanks for giving me the possibility of reviewing the manuscript entitled "Sleep and safety among health care workers: the role of OSA and sleep deprivation on safety" after the authors provided their updated review.
The current version of the manuscript significantly benefitted from the latest revision. It is now well organized, the data are provided in a logic flow, and it is a substantial contribution to the literature that looks at the detrimental consequences of OSA and sleep deprivation in healthcare workers.
I here identified only minor comments to be addressed, but rather than these minor corrections I deem the manuscript suitable for publication in the Medicina journal.
Minor corrections:
- in the abstract, in line 12 and in line 15 replace "patients" with "individuals" because you are talking on a general level (and also including doctors)
- line 19: "reviews" instead of "review"
- in the introduction: replace "percent" with % in line 26
- move the sentence of line 45-46 on the prevalence of OSA among HCW after line 36. Also, despite being sparse, provide an approximate prevalence anyway, as you are citing it
- Review of Literature:
- line 48: typo in "sleep morbidity"
- line 79: missing some part of the sentence
- line 86: what is "bidirectional"? OSA and what? It is not clear. Please, specify in the text
Author Response
We appreciate the reviewer’s positive feedback as well as vigilance to identify minor issues.
All the issues and typo mentioned above has been corrected and is in the document as track changes.
The last comment in line 86 has been revised for clarity as suggested.
We like to thank the reviewer for his time and patience
This manuscript is a resubmission of an earlier submission. The following is a list of the peer review reports and author responses from that submission.
Round 1
Reviewer 1 Report
Shaik and colleagues, in the manuscript entitled “Obstructive Sleep Apnea and Safety” described the relationship between OSA, its symptoms, and their effect on safety in the healthcare system.
Comments:
- The title should be rephrased as it is not clear – possibly. “Obstructive Sleep Apnea and Safety in Healthcare System”
- In the introduction, at least one sentence of the definition of OSA should be added.
- The prevalence of OSA stated in the manuscript is understated, please see e.g. doi: 10.1016/j.smrv.2016.07.002, 10.1016/S2213-2600(15)00043-0, 10.1016/S2213-2600(19)30198-5
- Several abbreviations in the manuscript are not expanded the first time they appear in the article, for example HCW.
- The aim of the study at the end of the introduction should be more detailed whether it discusses sleep disturbances in general or OSA in particular.
- While providing OSA comorbidities and its risk factors consider expanding literature (e.g. doi:10.1161/CIR.0000000000000988, 10.3390/jcm10173770 and doi: 10.1186/s12931-017-0616-8, 10.5664/jcsm.7166 respectively)
- While describing the consequences of OSA, please add how through circadian disturbances the sleep quality in this group is further worsening (e.g. doi: 10.3390/diagnostics11061082, 10.3390/jcm8101634, 10.3390/ijms23020709)as well as it accelerates the aging process (e.g. doi: 10.1038/s41598-020-63761-7. 10.1016/j.resp.2010.09.001, 10.3390/jcm9051599).
- The problem with long term CPAP adherence should be more highlighted (e.g. doi: 10.1016/j.sleep.2019.01.004, 10.3390/jcm11010139, 10.1186/s40463-016-0156-0).
- If the manuscript is to summarize both OSA and other sleep disorders please state that clearly in the title, abstract, and aim of the study.
Author Response
Shaik and colleagues, in the manuscript entitled “Obstructive Sleep Apnea and Safety” described the relationship between OSA, its symptoms, and their effect on safety in the healthcare system.
Comments:
- The title should be rephrased as it is not clear – possibly. “Obstructive Sleep Apnea and Safety in Healthcare System”
We appreciate the reviewer’s comment and the title have been changed to reflect it more appropriately.
- In the introduction, at least one sentence of the definition of OSA should be added.-
- We thank the reviewer for their comment and agree. We have added the comment
-“Obstructive sleep apnea is a sleep-related breathing disorder that causes repetitive partial or complete obstruction of upper airways during sleep”.
- The prevalence of OSA stated in the manuscript is understated, please see e.g. doi: 10.1016/j.smrv.2016.07.002, 10.1016/S2213-2600(15)00043-0, 10.1016/S2213-2600(19)30198-5
Rewritten and made changes to references:
Lechat B, Naik G, Reynolds A, Aishah A, Scott H, Loffler KA, Vakulin A, Escourrou P, McEvoy RD, Adams RJ, Catcheside PG, Eckert DJ. Multinight Prevalence, Variability, and Diagnostic Misclassification of Obstructive Sleep Apnea. Am J Respir Crit Care Med. 2022 Mar 1;205(5):563-569. doi: 10.1164/rccm.202107-1761OC. PMID: 34904935; PMCID: PMC8906484.
- Several abbreviations in the manuscript are not expanded the first time they appear in the article, for example, HCW. We appreciate the reviewer’s vigilance and have corrected those.
- The aim of the study at the end of the introduction should be more detailed whether it discusses sleep disturbances in general or OSA in particular.
-We appreciate the reviewer’s comment. The review was to highlight the importance of undiagnosed and untreated OSA and its safety consequence among healthcare workers. The airline and transportation industry has been way ahead of the curve and has strict requirements and screening methods in place and if OSA is to be able to work, it must show efficacy and compliance with the therapy. The Healthcare industry so far has not adopted and with long duty hours, it can have safety concerns.
- While providing OSA comorbidities and its risk factors consider expanding literature (e.g. doi:10.1161/CIR.0000000000000988, 10.3390/jcm10173770 and doi: 10.1186/s12931-017-0616-8, 10.5664/jcsm.7166 respectively)- Already discussed various risk factors.
- While describing the consequences of OSA, please add how through circadian disturbances the sleep quality in this group is further worsening (e.g. doi: 10.3390/diagnostics11061082, 10.3390/jcm8101634, 10.3390/ijms23020709) as well as it accelerates the aging process (e.g. doi: 10.1038/s41598-020-63761-7. 10.1016/j.resp.2010.09.001, 10.3390/jcm9051599). - Circadian disturbances and sleep quality effects on HCW have been discussed under Impact on HCW and Impact on patient care
- The problem with long term CPAP adherence should be more highlighted (e.g. doi: 10.1016/j.sleep.2019.01.004, 10.3390/jcm11010139, 10.1186/s40463-016-0156-0).
We agree with the reviewer's comment. CPAP adherence and compliance are an integral part. Discussing circadian rhythm would be beyond the scope of this review. We however did mention in our implications that regulations need to be established for assessing OSA control.
- If the manuscript is to summarize both OSA and other sleep disorders please state that clearly in the title, abstract, and aim of the study.
We appreciate the reviewer's comment and have connected the sleep issue with sleep deprivation to clarify and dissipate any confusion.

Reviewer 2 Report
line 10 OSA [OSA] repetition
line 22 OSA Explain the acronymous
reference 3 these data are very old and completely outdated after HypnoLaus Study
Heinzer R, Vat S, Marques-Vidal P, Marti-Soler H, Andries D, Tobback N, Mooser V, Preisig M, Malhotra A, Waeber G, Vollenweider P, Tafti M, Haba-Rubio J. Prevalence of sleep-disordered breathing in the general population: the HypnoLaus study. Lancet Respir Med. 2015 Apr;3(4):310-8.
The aim of the study is not clearly expressed.
It's necessary a description of how you found sources and how you analyzed them for inclusion and discussion in the review.
Author Response
line 10 OSA [OSA] repetition- edited
line 22 OSA Explain the acronymous done
reference 3 these data are very old and completely outdated after HypnoLaus Study changed to latest 2021 values
- Lechat B, Naik G, Reynolds A, Aishah A, Scott H, Loffler KA, Vakulin A, Escourrou P, McEvoy RD, Adams RJ, Catcheside PG, Eckert DJ. Multinight Prevalence, Variability, and Diagnostic Misclassification of Obstructive Sleep Apnea. Am J Respir Crit Care Med. 2022 Mar 1;205(5):563-569. doi: 10.1164/rccm.202107-1761OC. PMID: 34904935; PMCID: PMC8906484.
The aim of the study is not clearly expressed.
The aim of this study is to highlight the impact of OSA on HCW and to translate these effects into safety in healthcare. We have also addressed the importance and aim that we wanted to convey, as responded in the reviewer 1 comments. The review was to highlight the importance of undiagnosed and untreated OSA and its safety consequence among healthcare workers. The airline and transportation industry has been way ahead of the curve and has strict requirements and screening methods in place and if OSA is to be able to work, it must show efficacy and compliance with the therapy. The Healthcare industry so far has not adopted and with long duty hours it can have safety concerns
It's necessary to have a description of how you found sources and how you analyzed them for inclusion and discussion in the review.> This is not a systematic review; it is therefore not necessary. We appreciate the reviewer's vigilance in this regard.

Reviewer 3 Report
The review focus on the effect of OSA, specifically on the working environment of healthcare workers. It presents the consequences that uncontrolled OSA symptoms can have on healthcare, and patients. Then, a final paragraph shows that the healthcare workers are not the only one where OSA can have a detrimental effect.
The review presents data already known and it does not provide any original insight. Moreover, I believe that the major flaw is based on the inclusion of articles that talk of sleep deprivation that is not clearly caused by OSA. I think that the review, as the title is OSA and Safety, should only present consequences of OSA. Sleep deprivation when it is driven by long shift work or stress, is not a result of OSA anymore. Unless the author changes the topic and also include sleep deprivation and other factors in the title, otherwise it is scientifically logical that the paper should only include studies on OSA individuals and its consequences.
I believe that this constitutes a major flaw for the understanding of the review.
Below my extensive minor comments.
Abstract:
Line 10: Please, spell the acronym of OSA the first time you mention it.
Introduction:
Line 25-26: the concept is repeated twice.
Line 27: in lines 25-26, a figure of prevalence is provided. However, in line 27, the authors indicate “this pattern of incidence”. Prevalence and incidence are different concepts and should not be interchangeable. I believe the figure provided by the authors in the previous lines is referred to the prevalence, so the authors should be consistent with the terminology.
Line 22: Spell out OSA as it is the first time that is encountered in the manuscript. I would also provide a brief definition of OSA.
Line 30: did the authors want to indicate EDS instead of ESS? ESS is Epworth Sleepiness Scale, not excessive daytime sleepiness.
Line 31: HCW needs to be spelt out again (and also SD in line 32), despite presenting the abbreviation in the abstract.
Line 31: the sentence is confusing. Please, rephrase. It is not clear if you are talking about HCW that are also patients. IF that is the case, I suggest rephrasing like “when a poorly controlled OSA is seen in HCW, it can adversely affect …”
Line 34: as you mentioned that several studies report these findings, the sentence needs a reference.
Review of the literature
Lines 41-43: the authors need to provide references for this statement. Also, rather than the body weight, is the BMI that is the strongest risk factor.
Lines 50-51: needs a reference.
Line 53-55: needs to be completely reformulate. The statement is confusing and misleading. I understand that shift schedule may predispose to stress, but how shift schedule may predispose to “also the above-mentioned risk factors of OSA, including smoking, alchohol, inactivity”? How does shift schedule predispose to this? I also do not completely agree with the sentence where stress is a risk factor of OSA (lines 50-51), unless the authors provide a strong evidence. The paragraph needs to be re-structuded: the authors initially talk about stress, then mention inactivity, then they go back to talk about stress, then they talk about sedentary lifestyle.
Lines 60-63: I think that the authors should be more cautious to express this statement. At the best of my knowledge, the fact that HCW is at higher risk for OSA than the general population has never been proven, and I don’t see why this should be the case, as the strongest risk factors are not stress-related, but are the ones mentioned at the beginning of the section in lines 41-45.
Line 67: the acronym of SD has already been used before. No need to repeat it again.
In the section of “Impact on Sleep duration and Quality”, the topic is focused on sleep deprivation and insufficient sleep as cause of worsening of OSA. To my knowledge, although this may be true and may constitute a vicious cycle, what is more supported is the opposite, that is OSA is responsible of poor sleep quality and frequent awakenings. I would either rework the all paragraph, with this as major relationship, or at least introduce also this new element to support that poor sleep quality and sleep deprivation and OSA are indeed correlated (but not only in the direction expressed here by the authors).
Line 91: what does it mean that AHI, poor sleep quality, SD are the major challenge in OSA? They are characteristic features, not “challenge”.
Line 92 : EDS has already been presented as acronym. No need to repeat it here.
How does dry mouth result from symptoms like AHI, poor sleep quality and SD? Please, provide reference and explain.
Line 93-96: needs reference
Line 96-97: needs reference
Line 99: here you provide a percentage of depression (50%). however, a percentage with no comparison does not provide much sense. I suggest proposing the prevalence by comparing it to the depression of the general population. Also, the authors need to provide a reference for the statement.
Line 100: there are many other treatment for OSA beside the CPAP. If the authors talks about CPAP, at least report that it is the gold standard, which justify why the authors only address the use of the CPAP.
“Impact on HCW” section
Some of the studies reported to support evidence are not on OSA, but only on burnout, and long work shift. I believe only those where the participants did actually suffer from OSA should be report, and dismissing those studies where the individuals do not suffer from OSA.
“Impact on patient care” section
Also in this paragraph, the authors should just present those situation of SD driven by OSA, and dismiss the studies where SD and fatigue and daytime somnolence were given by long hours, stress, burnout. The article is on the effect of OSA in the performance and safety, not on the effect of sleep. And if sleep is impaired due to other consequences, OSA does not have any role in it. Otherwise the article needs to be retitle, with OSA and effect of sleep deprivation.
Line 176: here the authors use “obstructive sleep apnea”, spelt out. Acronym needs to be used in a consistent manner.
Table 1 needs an explicative legend. Remove the automatic text of the table.
Also, the table 1, which reflects the section on Impact on HCW, is based on the consequences of sleep deprivation. However, the title of the paragraph is OSA in HCW is a safety concern for all. Therefore, only studies with participants with OSA should be retained in the table. Indeed, these individuals might have sleep deprivation because of the exhaustive long work shift, the need of the personnel, which has nothing to do on OSA.
Author Response
The review focus on the effect of OSA, specifically on the working environment of healthcare workers. It presents the consequences that uncontrolled OSA symptoms can have on healthcare and patients. Then, a final paragraph shows that the healthcare workers are not the only ones where OSA can have a detrimental effect.
The review presents data already known and it does not provide any original insight. Moreover, I believe that the major flaw is based on the inclusion of articles that talk of sleep deprivation that is not clearly caused by OSA. I think that the review, as the title is OSA and Safety, should only present the consequences of OSA. Sleep deprivation when it is driven by long shift work or stress is not a result of OSA anymore. Unless the author changes the topic and also includes sleep deprivation and other factors in the title, otherwise it is scientifically logical that the paper should only include studies on OSA individuals and its consequences.
I believe that this constitutes a major flaw in the understanding of the review.
Below are my extensive minor comments.
We appreciate the reviewer's comment which is very valid. The title has been modified to reflect and clarify some changes. As mentioned earlier, the healthcare industry as compared to the airline and transportation industry has not required screening, diagnosis, and therapeutic efficacy requirements, the data is scarce. This article helps to bring about an important area for discussion that which healthcare industry need to consider. Ostrich's policy of burying the head in the sand will not help and healthcare professionals and regulators need to face this challenge head-on and address this important safety issue.
Abstract:
Line 10: Please, spell the acronym of OSA the first time you mention it. -done
Introduction:
Line 25-26: the concept is repeated twice. edited
Line 27: in lines 25-26, a figure of prevalence is provided. However, in line 27, the authors indicate “this pattern of incidence”. Prevalence and incidence are different concepts and should not be interchangeable. I believe the figure provided by the authors in the previous lines is referred to the prevalence, so the authors should be consistent with the terminology. Changed to prevalence.
Line 22: Spell out OSA as it is the first time that is encountered in the manuscript. I would also provide a brief definition of OSA. done
Line 30: did the authors want to indicate EDS instead of ESS? Corrected EDS: excessive daytime symptoms as clearly stated the first time used.
Line 31: HCW needs to be spelled out again (and also SD in line 32), despite presenting the abbreviation in the abstract. Done
Line 31: the sentence is confusing. Please, rephrase. It is not clear if you are talking about HCWs that are also patients. If that is the case, I suggest rephrasing like “when a poorly controlled OSA is seen in HCW, it can adversely affect …” -corrected
Line 34: as you mentioned that several studies report these findings, the sentence needs a reference. References were wrongly placed> corrected now.
Review of the literature
Lines 41-43: the authors need to provide references for this statement. Also, rather than the body weight, the BMI that is the strongest risk factor. - changed
Lines 50-51: needs a reference. given
Line 53-55: needs to be completely reformulated. The statement is confusing and misleading. I understand that shift schedule may predispose to stress, but how shift schedule may predispose to “also the above-mentioned risk factors of OSA, including smoking, alcohol, inactivity”? How does the shift schedule predispose to this? I also do not completely agree with the sentence where stress is a risk factor of OSA (lines 50-51), unless the authors provide strong evidence. The paragraph needs to be re-structured: the authors initially talk about stress, then mention inactivity, then they go back to talk about stress, then they talk about a sedentary lifestyle. Rewritten the paragraph to include references.
Lines 60-63: I think that the authors should be more cautious to express this statement. To the best of my knowledge, the fact that HCW is at higher risk for OSA than the general population has never been proven, and I don’t see why this should be the case, as the strongest risk factors are not stress-related, but are the ones mentioned at the beginning of the section in lines 41-45.
We appreciate the reviewer's comments. There is obviously a lack of data, but OSA's high prevalence accompanied by sleep deprivation among healthcare workers can increase the safety risk.
Line 67: the acronym of SD has already been used before. No need to repeat it again. Done
In the section “Impact on Sleep duration and Quality”, the topic is focused on sleep deprivation and insufficient sleep as causes of the worsening of OSA. To my knowledge, although this may be true and may constitute a vicious cycle, what is more, supported is the opposite, that is OSA is responsible for poor sleep quality and frequent awakenings. I would either rework the all paragraph, with this as a major relationship, or at least introduce also this new element to support that poor sleep quality and sleep deprivation and OSA are indeed correlated (but not only in the direction expressed here by the authors). rewritten
Line 91: what does it mean that AHI, poor sleep quality, and SD are the major challenge in OSA? They are characteristic features, not “challenge”.-rewritten
Line 92 : EDS has already been presented as an acronym. No need to repeat it here.
How does dry mouth result from symptoms like AHI, poor sleep quality, and SD? Please, provide a reference and explain. explained

Round 2
Reviewer 1 Report
The aim of the study should be rephrased as said before - not all sleep disorders are taken under consideration in the manuscript - please rephrase to show the focus on OSA.
Other comments have not been addressed well enough. The literature was not expanded to provide a more in-depth view on the topic - consider earlier suggestions.
Even basic suggestions such as providing expansion first before using abbreviation have not been included with needed attention - e.g, CPAP is not expanded the first time it appears in text - it is expanded the second time.
Author Response
Comment: The aim of the study should be rephrased as said before - not all sleep disorders are taken under consideration in the manuscript - please rephrase to show the focus on OSA.
Response: We appreciate the reviewer comments. We have focused mainly of Sleep deprivation and OSA among HCW and safety. Title of the manuscript has also been revised earlier as per suggestion by the other reviewers too.
Other comments have not been addressed well enough. The literature was not expanded to provide a more in-depth view on the topic - consider earlier suggestions.
Response: We appreciate the reviewer concern. We have added more reviews throughout the manuscript and have added several new and updated references too. We agree with their comments as it helped to improve our manuscript. We respectfully, again mention that this is not a systematic review, so literature search criteria which applies to systematic review does not apply here. We though, appreciate the reviewer vigilance and concern
Even basic suggestions such as providing expansion first before using abbreviation have not been included with needed attention - e.g, CPAP is not expanded the first time it appears in text - it is expanded the second time.
Response: We apologize for our oversight in this regard and thank the reviewer for their vigilance. This has been corrected
Reviewer 3 Report
The authors have tried their best to address my previous concerns. By changing the title and including "sleep deprivation", all the arguments and topics fit into the manuscript.
Only a minor oversight:
In the abstract, sleep deprivation is named as SD in line 20, with no explanation of the acronym earlier in the abstract.
Author Response
Comments: The authors have tried their best to address my previous concerns. By changing the title and including "sleep deprivation", all the arguments and topics fit into the manuscript.
Response: Thank you
Only a minor oversight:
In the abstract, sleep deprivation is named as SD in line 20, with no explanation of the acronym earlier in the abstract.
Response: We apologize for our oversight and has been corrected.